# A Flexible Supercapacitor Based on Niobium Carbide MXene and Sodium Anthraquinone-2-Sulfonate Composite Electrode

**DOI:** 10.3390/mi14081515

**Published:** 2023-07-28

**Authors:** Guixia Wang, Zhuo Yang, Xinyue Nie, Min Wang, Xianming Liu

**Affiliations:** 1Henan Key Laboratory of Function-Oriented Porous Materials, College of Chemistry and Chemical Engineering, Luoyang Normal University, Luoyang 471934, China; 2School of Pharmaceutical Sciences, Chongqing University, Chongqing 401331, China

**Keywords:** niobium carbide, MXene, sodium anthraquinone-2-sulfonate, supercapacitors, energy storage, composite electrode

## Abstract

MXene-based composites have been widely used in electric energy storage device. As a member of MXene, niobium carbide (Nb_2_C) is a good electrode candidate for energy storage because of its high specific surface area and electronic conductivity. However, a pure Nb_2_C MXene electrode exhibits limited supercapacitive performance due to its easy stacking. Herein, sodium anthraquinone-2-sulfonate (AQS) with high redox reactivity was employed as a tailor to enhance the accessibility of ions and electrolyte and enhance the capacitance performance of Nb_2_C MXene. The resulting Nb_2_C–AQS composite had three-dimensional porous layered structures. The supercapacitors (SCs) based on the Nb_2_C–AQS composite exhibited a considerably higher electrochemical capacitance (36.3 mF cm^−2^) than the pure Nb_2_C electrode (16.8 mF cm^−2^) at a scan rate of 20 mV s^−1^. The SCs also exhibited excellent flexibility as deduced from the almost unchanged capacitance values after being subjected to bending. A capacitance retention of 99.5% after 600 cycles was observed for the resulting SCs, indicating their good cycling stability. This work proposes a surface modification method for Nb_2_C MXene and facilitates the development of high-performance SCs.

## 1. Introduction

With the increasing demand for electric energy, electric energy storage has attracted increasing attention from researchers [1,2,3]. There are several electric energy storage sources, including aqueous Zn–ion batteries [4,5], lithium–selenium batteries [6], Li–ion batteries [7,8], Zn–air batteries [9], ammonium-ion batteries [10], and supercapacitors (SCs) [11,12,13]. Among them, SCs possess useful characteristics such as ultrahigh power density, long lifetime, and environmental sustainability, and thus, they have become promising electrical energy storage materials for portable electronics and other electric devices [14]. SCs can be divided into electrical double-layer capacitors (EDLCs) and pseudocapacitors in accordance with the energy storage mechanism. Compared with EDLCs, pseudocapacitors usually exhibit much higher capacitances and energy densities through Faradaic redox reactions [1,15]. Pseudocapacitive materials include conducting polymers, electrochemically active organic molecules, and transition metal compounds. As the representative electrochemically active organic molecules, quinones and their derivatives exhibit high redox reactivity, high capacitance, and good adjustable electrochemical reversibility, and they have been used as electrode materials of SCs [16,17,18,19,20,21]. However, the capacitive performance and long-term cycle stability of SCs based on organic quinones or their derivatives are relatively low due to their low electronic conductivity. To enhance the electronic conductivity of quinones and their derivatives, the Shi group [22] and Zhu group [23] proposed a strategy of using reduced graphene oxide (rGO) as a conductive support of sodium anthraquinone-2-sulfonate (AQS). The resulting SCs based on AQS–rGO composites had enhanced capacitive performance and high capacity retention.

MXenes are a new family of multifunctional two-dimensional (2D) materials comprising transition metal carbides, nitrides, and carbonitrides [24,25]. Titanium carbide MXene-based nanocomposites exhibit good structural stability, electrical properties, and electrocatalytic activity, which makes them very suitable for good matrices of SCs. Niobium carbide (Nb_2_C) MXene has a similarly high electronic conductivity to that of titanium carbide, excellent dispersibility in various solvents, and good compatibility with various components. Recently, Nb_2_C MXene has been revealed to be a good electrode candidate for energy storage due to its high specific surface area and high conductivity [26,27,28]. Both single Nb_2_C MXene (denoted as Nb_2_CT_x_) [29] as well as composites consisting of Nb_2_C MXene delaminated with isopropylamine with carbon nanotubes [30] can be used as electrode materials with excellent cycle stability and Li capacities.

Herein, the organic active molecule AQS was used as a tailor to modify 2D Nb_2_C MXene and the resulting Nb_2_C–AQS composite was employed to construct a novel SC with high capacitance. The SO_3_^−^ functional group in AQS can render AQS soluble in aqueous solutions as well as facilitate the combination of AQS and Nb_2_C MXene on the molecular level. Then, the electrochemical capacitance and cycle stability of the SC based on the as-prepared Nb_2_C–AQS composite were evaluated and also compared with bare Nb_2_C MXene.

## 2. Materials and Methods

### 2.1. Materials and Reagents

Nb_2_AlC powder was purchased from Forsman Scientific (Beijing, China) Co., Ltd. Hydrofluoric acid (HF, 40%) was from Tianjin Deen Chemical Reagents Co., Ltd., Tianjin, China. Tetrapropylammonium hydroxide (TPAOH, 25 wt%) was from J&K Scientific Co., Ltd., Beijing, China. Sodium anthraquinone-2-sulfonate (AQS, 99%) was purchased from Sigma–Aldrich (St. Louis, MO, USA). Sodium sulfate and other chemicals were all reagent grade quality or better and used without further purification. Aqueous solutions were prepared from deionized water (>18 M·cm, Milli-Q purification system).

### 2.2. Methods

#### 2.2.1. Synthesis of Nb_2_C MXene

Two-dimensional Nb_2_C MXene was prepared by a modified chemical exfoliation method [28]. Nb_2_AlC powder (ca. 5 g) was slowly added to 30 mL of 40% aqueous HF (Note: HF is highly corrosive) and stirred for 20 h at room temperature. After collection by centrifugation (3500 rpm) and washing with water and anhydrous ethanol, the precipitate was dispersed in 30 mL of aqueous tetrapropylammonium hydroxide under stirring for 10 h at room temperature. The raw Nb_2_C materials were collected by centrifugation and washed 3× with ethanol and water to remove the residual TPAOH. Then, the precipitate was dispersed in 100 mL of deionized water and sonicated at 200 W for 30 min. The supernatant was collected and dried in a vacuum oven at 80 °C for 6 h. Thus, the resulting Nb_2_C MXene was obtained.

#### 2.2.2. Preparation of Nb_2_C–AQS Composite

The Nb_2_C–AQS composite was prepared by a simple one-step hydrothermal method. Aqueous Nb_2_C and AQS (30 mL) with a mass ratio of 1:3 was stirred for 6 h at room temperature. Then, the mixture was transferred into a 50-mL Teflon-lined stainless-steel autoclave and heated at 180 °C for 12 h. Nb_2_C–AQS composite was obtained after washing the aforementioned Nb_2_C–AQS solution with deionized water 2× and freeze-drying under vacuum.

#### 2.2.3. Characterization

Scanning electron microscopy (SEM) images were acquired on a field-emission Sigma 500 microscope (Carl Zeiss, Jena, Germany). X-ray diffraction (XRD) was measured with a D8 Advance (Bruker, Mannheim, Germany) system. Fourier-transform infrared spectra (FTIR) were obtained with a Nicolet 6700 spectrometer (Thermo Nicolet, Madison, WI, USA).

The electrochemical performance of as-prepared Nb_2_C–AQS was carried out with a three-electrode system in 0.1 mol L^−1^ Na_2_SO_4_, with platinum foil as the counter electrode and Ag/AgCl as the reference electrode. The working electrode was fabricated as follows: the as-prepared active material (40 mg), acetylene black, and polytetrafluoroethylene were first mixed in a mass ratio of 75:15:10 in a small quantity of absolute ethanol in a manner that formed a homogeneous slurry. Then, the slurry was coated on a piece of nickel foam (1.0 cm × 1.0 cm), which was dried in a vacuum oven at 80 °C for 12 h and pressed before measurement. Cyclic voltammetry (CV) curves and galvanostatic charge/discharge (GCD) were all performed with a CHI 660E electrochemical workstation (CH Instruments).

## 3. Results and Discussion

### 3.1. Synthesis and Characterization of Nb_2_C–AQS Composite

Figure 1 illustrate the synthesis process of the Nb_2_C–AQS composite. The morphology of Nb_2_C and the corresponding Nb_2_C–AQS composite was characterized by SEM. Nb_2_C has an accordion-like lamellar structure similar to that of Ti_3_C_2_ MXene [24,31], and its layered structure has a large lateral size with almost no defects (Figure 2a). Compared with Nb_2_C, the Nb_2_C–AQS composite maintained the layered structure of Nb_2_C, but it has 3D porous structures (Figure 2b), attributable to the decorated AQS molecules as spacers that suppress aggregation of Nb_2_C MXene. The porous structures of the Nb_2_C–AQS composite can act as ion buffers that facilitate ion transport between the Nb_2_C layers. The exiting of the S element also indicates the successfully assembly of AQS molecules on Nb_2_C MXene surface (Figure 2d–h).

### 3.2. Chemical Structure

The XRD patterns indicate formation of Nb_2_C materials (Appendix A). Only one intense peak of Nb_2_C at ca. 25° was observed, and most of the non-basal plane peaks of Nb_2_AlC bulk (red line in Appendix A), including the most intense diffraction peak at ca. 39°, were essentially no longer evident after HF treatment and TPAOH intercalation (black line in Appendix A). Compared with the diffraction patterns of AQS (blue line in Figure 3a), the Nb_2_C–AQS composite had similar XRD patterns, but the peak intensities all decreased, due to interaction of Nb_2_C MXene and AQS molecules (black line in Figure 3a). The FTIR spectra of the Nb_2_C–AQS composite exhibited characteristic peaks of AQS at 1215 and 1660 cm^−1^ (Figure 3b), attributable to the asymmetric stretching vibration of –SO_3_^−^ and the stretching vibration of –C=O from AQS, respectively [32,33]. The zeta potential of the Nb_2_C–AQS composite was −31.8 mV, a much more negative value than that of the Nb_2_C films (−0.19 mV). The results strongly demonstrate the anchoring of AQS onto Nb_2_C MXene.

### 3.3. Electrochemical Performance

The obtained Nb_2_C–AQS composite had good conductivity with an equivalent-series resistance (ESR) value of ca. 0.5 Ω derived from its Nyquist plots (Appendix A). The CV profiles of the Nb_2_C electrode exhibited a nearly rectangular shape at low scan rates in the range of 10–100 mV s^−1^ (Figure 4a), which is characteristic of EDLC behavior. Compared with bare Nb_2_C, the resultant device based on the Nb_2_C–AQS composite (denoted as-prepared) exhibited both Faradaic and EDLC capacitance with a pronounced single pair of peaks at ca. 0.22/0.45 V versus Ag/AgCl (at 20 mV s^−1^), originating from the redox reactions of the AQS molecules [22] (Figure 4b). Moreover, the CV curves of Nb_2_C–AQS remained well-defined and symmetric even at a high scan rate of 100 mV s^−1^, suggesting excellent reversibility and stability of the Nb_2_C–AQS electrode. Notably, the peak potentials of the redox peaks slightly shifted with increasing scan rate, which might be due to partial proton-coupled charge transfer between AQS and Nb_2_C. Circa 70% of the capacitance was retained when the scan rate was increased to 100 mV s^−1^ (25.4 mF cm^−2^; refer to the Appendix A for the calculation methods of the specific capacitance), indicating its good rate capability. The Nb_2_C–AQS composite electrode exhibited a considerably higher electrochemical capacitance (36.3 mF cm^−2^ at 20 mV s^−1^) than the bare Nb_2_C electrode (16.8 mF cm^−2^ at 20 mV s^−1^) (Figure 4c, Table 1); this is mainly due to the high pseudocapacitance of AQS as well as the enhanced accessibility of ions and electrolyte, facilitated by the unique porous structure of the Nb_2_C–AQS composite. The 42.7 mF cm^−2^ of specific capacitance for the resulting Nb_2_C–AQS capacitor is higher than that of graphene-based micro-supercapacitors (micro-SCs) (Table 2). The as-prepared SCs display similar capacitance performance as those in some other published works [34,35,36,37,38,39,40,41].

The CVs of the SCs exhibited similar shapes to that of as-prepared SCs when subjected to bending at a scan rate of 10 mV s^−1^ (Figure 4d). The specific capacitances were 48.1 and 40.0 mF cm^−2^ for concavely bent and convexly bent SCs (Table 1, bending modes in Appendix A), respectively, which are almost the same as that of as-prepared SCs (42.7 mF cm^−2^). Thus, the as-prepared SCs have excellent flexibility.

The GCD curves of the Nb_2_C–AQS composite included two parts (Figure 4e): a pair of charge–discharge plateaus attributable to the anchored AQS, and an oblique line stage due to the EDLC behavior of Nb_2_C, which coincides well with the CV curves in Figure 3b. The specific capacitance of the resulting SCs was 36.0 mF cm^−2^ at a current density of 15 mA cm^−2^ (for the calculation of the specific capacitance, refer to the Appendix A). In addition, the device exhibited a capacitance retention of 99.5% of its initial capacitance after 600 cycles (Appendix A), indicating good cycling stability.

## 4. Conclusions

In summary, AQS was anchored successfully onto the surface of Nb_2_C MXene, and the obtained composite had good conductivity as well as a porous structure. SCs based on the Nb_2_C–AQS composite exhibited a high specific capacitance and good cycling stability. The SCs maintained their initial capacitance when subjected to bending, indicating good flexibility. Compared with bare Nb_2_C MXene, the Nb_2_C–AQS composite electrode provides higher capacitance, which was mainly ascribed to Faradaic redox reactions of the anchored AQS molecules as well as the enhanced accessibility of ions and electrolyte. This work proposes a surface functionalization method for Nb_2_C MXene and also provides a strategy for enhancing the capacitance performance of pure MXene, which will facilitate the development of high-performance SCs.

## 5. Patents

The results of the patent (a preparation method of a supercapacitor based on 2D niobium carbide nanocomposites, No. ZL2019109099814, China) are from the work reported in this manuscript.

## Figures and Tables

**Figure 1 micromachines-14-01515-f001:**
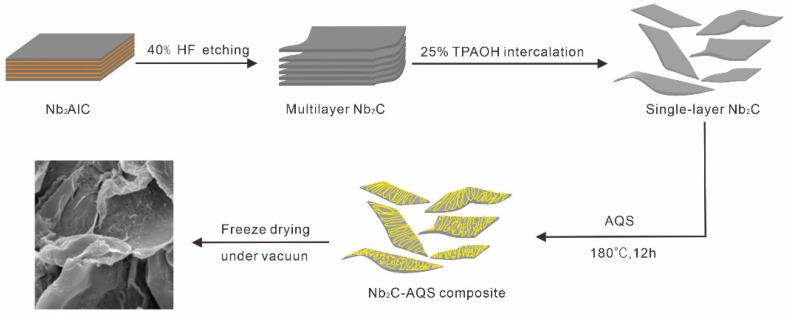
Schematic diagram for the fabrication of Nb_2_C–AQS composite.

**Figure 2 micromachines-14-01515-f002:**
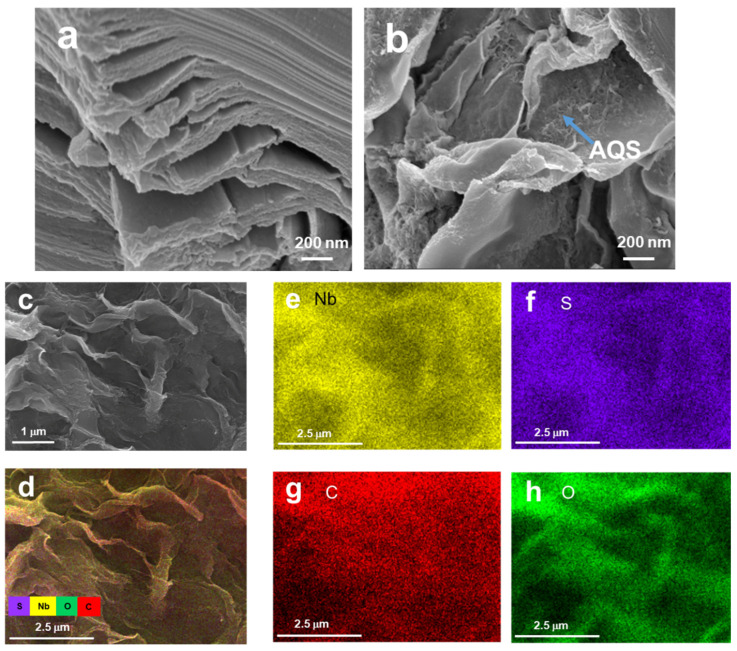
SEM images: (**a**) Nb_2_C layers; (**b**) and (**c**): Nb_2_C–AQS composite; (**d**–**h**) element mapping of (**c**).

**Figure 3 micromachines-14-01515-f003:**
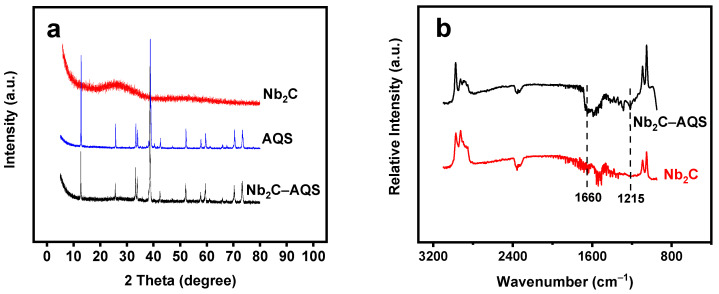
(**a**) XRD patterns of Nb_2_C–AQS, AQS, and Nb_2_C MXene. (**b**) FTIR spectra of Nb_2_C–AQS and Nb_2_C MXene.

**Figure 4 micromachines-14-01515-f004:**
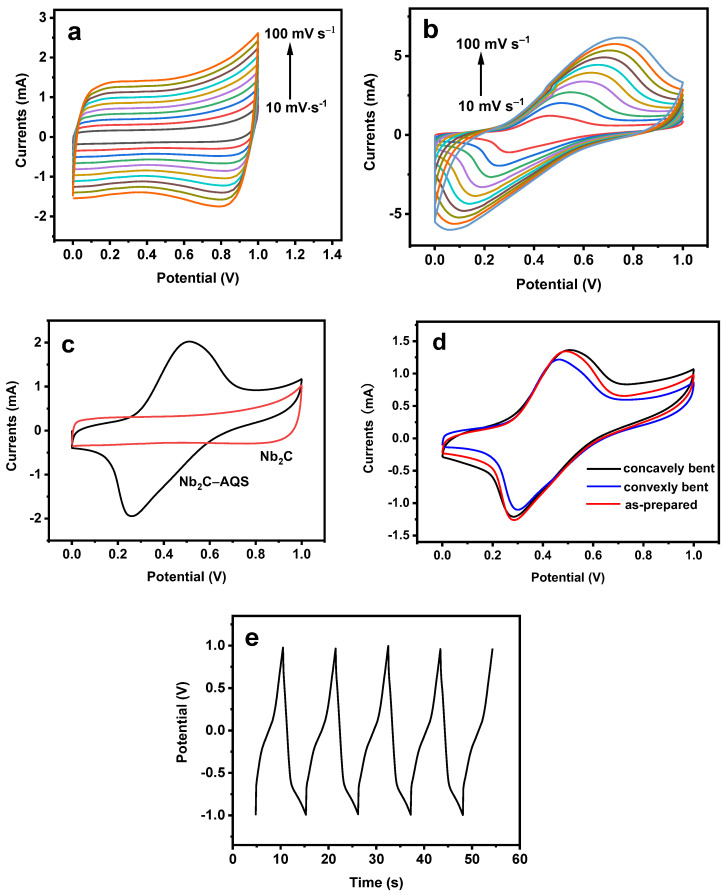
CV and GCD curves: (**a**) CV curves of Nb_2_C MXene and (**b**) Nb_2_C–AQS composite-based SCs at various scan rates. (**c**) CV curves of Nb_2_C MXene (red line) and Nb_2_C–AQS composite (black line) at a scan rate of 20 mV s^−1^. (**d**) CV curves of Nb_2_C–AQS-based SCs (as-prepared, red line), concavely bent (black line), and convexly bent (blue line) at a scan rate of 10 mV s^−1^. (**e**) GCD curves of the as-prepared SCs at a current density of 15 mA cm^−2^. All the electrochemical measurements were performed in 0.1 mol L^−1^ Na_2_SO_4_.

**Table 1 micromachines-14-01515-t001:** Electrochemical parameters for as-prepared MSCs in different conditions.

	Parameters	Areal Capacitance (mF cm^−2^)	Scan Rate
Electrode Materials			
Nb_2_C–AQS		36.3	20 mV s^−1^
Nb_2_C		16.8	20 mV s^−1^
Nb_2_C–AQS		42.7	10 mV s^−1^
Concavely bent		48.1	10 mV s^−1^
Convexly bent		40.0	10 mV s^−1^

**Table 2 micromachines-14-01515-t002:** Supercapacitance comparison of several composite electrode materials.

Electrode Materials	Specific Capacitance	Test Condition	Ref.
Nb_2_C–AQS	36.3 mF cm^−2^	20 mV s^−1^	This work
PET-CGO-LGO	0.756 mF cm^−2^	20 mV s^−1^	[42]
MPG	0.0807 mF cm^−2^	-	[43]
Ultrathin rGO	0.462 mF cm^−2^	0.1 μA g^−1^	[44]

## Data Availability

All data that support the findings of this study are available within the article and its Appendix A.

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
