# Peer review of "A Flexible Supercapacitor Based on Niobium Carbide MXene and Sodium Anthraquinone-2-Sulfonate Composite Electrode"

_micromachines, 2023, doi:10.3390/mi14081515_

Round 1

Reviewer 1 Report

The authors reported the facile synthesis of Nb2C carbide–sodium anthraquinone-2-sulfonate electrodes and the electrochemical properties of the composite in three-electrode systems were investigated. Generally speaking, the obtained results are normal and the capacitance of the electrode is limited. And there are some problems unsolved:

1. The baselines of the FTIR spectra in Fig.2 (b) are needed to be reconstructed.

2. The authors claimed that two charge storage mechanisms were involved in the charge storage process, and what redox reactions happened during the electrochemical process? Please give reasonable explanations.

3. Why the capacitance abruptly decreased after about the 600 cycles of charging-discharging tests in Fig. S4?

4. The composite electrode is expected to be further assembled into an asymmetric supercapacitor device with activated carbon electrode to investigate its application potential in the revised manuscript. 

please double check the whole manuscript

Reviewer 2 Report

In the presented Communication, the authors described a method to fabricate a MXene-based composite electrode material for supercapacitors, and characterized the chemical composition and the electrochemical behaviors of the produced composite materials.

There are several things can be improved in this article.

The title is not concise enough, and not well aligned with the body.

The abstract needs to be revised. It should focus on the composite material and its performances.

The changes in the morphology does not provide enough evidence for the existence of AQS. EDX results should be added along the SEM images.

Why did you switch from one electrolyte to another? What’s the point? The testing voltage range should be consistent. Which electrolyte was used in the rest of the electrical characterizations? Why different electrolytes were used for the ones in the supplementary?

In Figure S4, why the retention dropped dramatically around 600 cycles?

Compared to similar existing works, what is the advantages of the material produced in this Communication?

The authors managed to present the preparation of the composite material, and did reasonable characterizations. I would suggest that reconsider this article after major revision.

Moderate editing of English language required

Reviewer 3 Report

In this communication, aiming the energy storage, authors prepared 2D, niobium carbide mxene based, redox active composites as electrode. Further the supercapacitance performance was investigated. Comprehensive characterizations have been performed. In general, it is an interesting work and the manuscript is well organized. However, there are still some issues to be addressed. A moderate revision is suggested before its acceptance.

1.     More solid data should be provided in abstract section.

2.     One or two more keywords can be added.

3.     The generally introduction of the different energy storage sources should be provided with some more recent supporting articles, such as aqueous Zn-ion batteries (Journal of Alloys and Compounds, 2022, 903: 163824); lithium–selenium batteries (Rare Metals, 2022, 41(10): 3432-3445); Li-ion battery (New Journal of Chemistry 45, 19446-19455, 2021); Zn-air battery (Molecules 28 (5), 2147, 2023); supercapacitor (Journal of Bioresources and Bioproducts, 7, 4, 245-269, 2022); ammonium-ion battery (Chemical Engineering Journal, 2023, 458, 141381); etc.

4.     It is better to add one sub-section to introduce the raw materials.

5.     One scheme to show the experimental procedure is suggested for better understanding of this work to readers.

6.     It is better to rebuild the scale bar for better readability.

7.     The element mapping should be added.

8.     The sample name is already indicated in the curve, then the figure caption for fig. 2 can be simplified.

9.     The combination of Fig. 3 can be optimized.

10.  Conclusion is too short. More solid conclusions can be added.

11.  The section 3 is too short. More comparison and discussion can be added by making one table in the main manuscript for the supercapacitance comparison. Please consider the following articles: Journal of Energy Storage 67, 107559, 2023; High mass-loading α-Fe2O3 nanoparticles anchored on nitrogen-doped wood carbon for high-energy-density supercapacitor; Journal of Colloid and Interface Science 599, 443-452, 2021; Polymer 235, 124276, 2021; Polymers 14 (13), 2521, 2022; New Journal of Chemistry 45 (48), 22602-22609, 2021; Journal of Colloid and Interface Science 609, 179-187, 2022; Chinese Chemical Letters 31 (7), 1986-1990, 2020; etc.

12.  In order to demonstrate this journal is the right journal for your paper and also the connection of your paper to the composite community, it would be good to include some latest papers published in this journal as references.

13.  There are still some typos and grammar issues in the manuscript. Authors should carefully recheck the whole manuscript.

Round 2

Reviewer 1 Report

The composite electrode is needed to be further assembled into an asymmetric supercapacitor device  to investigate its application potential. 

double check the typo errors

Author Response

We are very thankful to the academic editor for his or her comments, we have reorganized the table 2 and recited related works with similar material and structure and reported unit.

Reviewer 2 Report

The reviewer's comments have been addressed. I agree to accept the manuscript for publication.

Author Response

(The authors gave the same response as above.)
